# A Comparison of Public Perceptions of Physicians and Veterinarians in the United States

**DOI:** 10.3390/vetsci7020050

**Published:** 2020-04-22

**Authors:** April A. Kedrowicz, Kenneth D. Royal

**Affiliations:** Department of Clinical Sciences, North Carolina State University, Raleigh, NC 27607, USA; kdroyal2@ncsu.edu

**Keywords:** veterinarians, physicians, public perceptions, veterinary profession

## Abstract

Veterinary medicine consists of virtually the same medical specialties as human medicine, with veterinarians performing similar roles as medical doctors, albeit with different species. Despite these similarities, anecdotally, some perceptions of veterinarians as not “real doctors” persist. The purpose of this study was to explore and compare public perceptions of physicians and veterinarians. Participants were provided brief definitions of a physician and a veterinarian and then asked to provide a rating that best describes their perception with respect to 25 different personality characteristics/traits. A sample of 606 participants (unweighted) in the United States completed the survey. The results of this research show that the public tends to perceive veterinarians more favorably than physicians. More specifically, veterinarians were viewed as more approachable, sensitive, sympathetic, patient and understanding, while physicians were viewed as more proud, arrogant and overconfident. These results point to the favorable public perceptions of veterinarians. These findings are particularly relevant for veterinary educators who train the future workforce and have a significant role both in how the profession is portrayed and emphasizing the relationship between the public trust and social responsibility. Reinforcing the public’s strong trust in the veterinary profession throughout students’ education could enhance their own self-concept, self-esteem and overall mental health and well-being.

## 1. Introduction

A 1999 study of the veterinary profession conducted in the United States found that veterinarians rate favorably among the pet owning public in comparison to other professionals, including physicians [1]. In particular, veterinarians ranked first in compassion, honesty and trustworthiness [1]. In 2006, Americans rated veterinarians as the third most honest and ethical professionals behind only nurses and pharmacists, and ahead of physicians [2]. However, patient visits to veterinary hospitals have been decreasing [3] and pet owners’ perceptions about veterinary hospitals have decreased since 2013 [4]. In particular, the percentage of pet owners who believe that their veterinary hospital provides staff appropriate training, adheres to the latest standards, has appropriate facilities and provides good value have decreased over a three-year period according to a recent the American Animal Hospital Association study [4]. 

A recent study of veterinary students’ values showed that the values that ranked highest for students were having fun in work and life, feeling respected for their work and gaining a sense of achievement, all intrinsic values [5]. Another noteworthy finding from this research was that veterinary students did not rate public contact high on the list of extrinsic values that were important to them, despite the fact that veterinarians will typically be called upon to be leaders in their community and thus, have a certain degree of contact with the public that will undoubtedly influence the public perceptions of the profession. Indeed, one aim of veterinary ethics teaching in the veterinary medicine curriculum stresses the importance of recognizing how their conduct can affect the public perception of veterinarians and how the development of appropriate personal and professional qualities can improve the public perception of the veterinary profession [6].

Veterinary medicine consists of virtually the same medical specialties as human medicine, with veterinarians performing similar roles as medical doctors, albeit with different species. Despite these similarities, anecdotally, some perceptions of veterinarians as not “real doctors” persist. Indeed, calls for enhancing veterinary medicine’s public image became prominent in the 1990s with attention dedicated to increasing the awareness about the roles, responsibilities and diverse activities of veterinarians, as well as the value of veterinary medicine to communities and the public [7]. Veterinarians play an important role in the One Health initiative, the interplay of human, animal and environmental health [8]. Collaboration between veterinarians, physicians, animal scientists and other public health experts is required to solve the complex problems related to human and animal health, food safety, zoonotic disease and antimicrobial resistance, to name a few. Nonetheless, further evidence has suggested that the public may be losing confidence in veterinarians, despite the 2006 Gallup study results. News and social media outlets portray pet owner concerns about costs of veterinary care and the speculation about veterinarians’ motives with respect to money and medical care [9,10], which ultimately have the potential to erode trust in the profession as a whole [11]. 

The purpose of this study was to understand the public’s perceptions of physicians and veterinarians to determine if differences exist. This is of particular interest since veterinarians have not been included in the Gallup Poll of the most honest and ethical professions survey since 2006. Results of this research will be useful for veterinary professionals to discern public perceptions of veterinarians. Furthermore, because the data were comparative to physicians, one of the most respected professions in the United States, it will help identify the extent to which veterinarians are perceived as “real doctors” by members of the public. Understanding how veterinarians are perceived by the public will help the profession to identify areas for which it wishes to alter or improve its image. These findings should be particularly relevant for veterinary educators who train the future workforce and have a significant role in how the profession is portrayed to future members, as well as the veterinary students and potential veterinary students who want to understand the current perceptions of the profession before formally becoming a member.

## 2. Materials and Methods

### 2.1. Participants

In order to attain a nationally representative sample within a margin-of-error comparable to US national public opinion polls we surveyed 606 participants. This resulted in a margin-of-error of 3.18% when using a 95% confidence level. In order to ensure that data were representative of the US population estimates, post-stratification statistical weights were utilized. More specifically, auxiliary statistics from the 2017 US Census data [12] served as the basis for generating statistical weights. Variables used in the weighting process included sex, age and race/ethnicity. Details of the process used to assign statistical weights were described in [13]. Upon the application of post-stratification weights, the data set consisted of 758 participant responses. A breakdown of unweighted demographic characteristics is presented in Table 1.

### 2.2. Instrumentation

Anderson (1968) generated a list of 555 personality trait words and asked college students to rate each in terms of favorability [14]. Anderson’s paper listed these traits in rank-order from most favorable to least favorable. We reviewed the list and chose 25 words (a combination of favorable, unfavorable and neutral words) that we believed would be particularly salient given a common mission of patient care for both physicians and veterinarians.

Participants were provided a brief definition of a physician and then asked to provide a rating that best described their perception with respect to 25 different characteristics/traits. Specifically, the statement read:


*Physicians, also known as “medical doctors” tend to the healthcare needs of humans. Although physicians may specialize in any number of medical specialties (e.g., pediatrics, surgery, pathology, etc.), we are interested in your general, overall perception of medical doctors. Please rate the degree to which you believe each characteristic below accurately describes physicians.*


After completing the items, participants proceeded to a second set of items in which they were provided a brief definition of a veterinarian and asked to again provide a rating that best described their perception with respect to 25 different characteristics/traits. Specifically, the statement read:


*Veterinarians, also known as “vets”, tend to the healthcare needs of animals, including pets, livestock, zoo and laboratory animals. Although veterinarians may specialize in any number of medical specialties (e.g., general practice, exotics, equine, etc.), we are interested in your general, overall perception of vets. Please rate the degree to which you believe each characteristic below accurately describes veterinarians.*


Importantly, each participant served as their own control. Once participants provided ratings for physicians they were unable to revisit the previous screen to identify their previous responses. The inability to refer to previous responses enabled us to treat both sets of responses as independent and mitigate any influence of previous answers, a term referred to as “statistical dependency” or “local item dependence” in the psychometrics literature.

With respect to the rating scale, we were concerned that a four- or five-point rating scale would result in a restriction of the range issue that would produce little statistical variation. Therefore, we opted to use a nine-point semantic differential scale with anchors to provide better discrimination among responses. More specifically, the rating scale consisted of the following: 1 (not at all), 2, 3 (a little), 4, 5 (somewhat), 6, 7 (mostly), 8, and 9 (completely).

### 2.3. Procedures

Data were collected in March 2019 using the Qualtrics survey software (Provo, UT). Participants were acquired using Amazon Mechanical Turk (mTurk) (Amazon.com, Inc. Seattle, WA, USA), a popular Internet crowdsourcing marketplace for hiring survey respondent “workers”. Participants were acquired from mTurk due to the previous research that has consistently shown that mTurk samples provided more culturally and socioeconomically diverse participants than other methods and the quality of the data was at least as good as the data collected via other survey platforms [15]. Inclusion criteria for the study required participants to be currently residing in the United States and at least 18 years old. Each mTurk worker was compensated for their participation. The study was declared ‘exempt’ by the university’s Institutional Review Board (NCSU IRB protocol #15528).

### 2.4. Analysis

SPSS statistical software (version 25, City, state abbre, USA) was used to perform all statistical analyses. In addition to descriptive statistics, paired-samples *t*-tests were performed to compare each individual’s responses to the common characteristic/trait using *p* < 0.05 as the criterion for statistical significance. Cohen’s *d* [16] effect sizes estimates were also calculated to determine the practical significance of any mean score differences. According to Cohen, values ≤ 0.20 indicate a small effect size, values approximating 0.50 indicate a moderate effect size and values ≥ 0.80 indicate a large effect size.

## 3. Results

The results indicated statistically significant differences for 24 of the 25 characteristics (see Table 2). The only characteristic with no statistically significant difference was ‘scientific’ with *p* = 0.583. The characteristics illustrating the largest effect sizes (*d* ≥ 0.800) included ‘approachable’, ‘proud’, ‘sensitive’, ‘arrogant’ and ‘sympathetic’. More specifically, veterinarians were perceived to be more approachable, sensitive and sympathetic, whereas physicians were perceived to be more proud and arrogant. 

Items with moderate–large effect sizes (0.667 ≤ *d* ≥ 0.799) included ‘punctual’, ‘overconfident’, ‘patient’ and ‘understanding’. More specifically, veterinarians were perceived to be more punctual, patient and understanding. Physicians, on the other hand, were perceived to be more overconfident. Items exhibiting moderate effect sizes (*d* = 0.500 to 0.666) included ‘caring’, ‘likable’, ‘helpful’, ‘respectful’ and ‘attentive’. More specifically, veterinarians were perceived to be more likable, helpful, respectful and attentive than physicians.

Items exhibiting low–moderate effect sizes (0.367 ≤ *d* ≥ 0.499) included ‘unpleasant’, ‘thorough’, ‘greedy’ and ‘ethical’. More specifically, veterinarians were perceived as more thorough and ethical and less greedy and unpleasant. Items exhibiting small effective sizes (*d* ≤ 0.366) included ‘honest’, ‘confident’, ‘inefficient’, ‘competent’, ‘rational’ and ‘skilled’. Veterinarians were perceived as more honest, competent, rational, and skilled, whereas physicians were perceived as more confident and inefficient.

## 4. Discussion

The results of this research showed that the public tends to perceive veterinarians more favorably than physicians. More specifically, veterinarians were viewed as more approachable, sensitive, sympathetic, patient and understanding, while physicians were viewed as more proud, arrogant and overconfident. We offer several possible explanations for these perceptions. First, it is possible that respondents’ personal experience with veterinarians, and/or others’ reported experience, influenced these results. Increasingly, empathy, communication and a collaborative approach to care are emphasized as part of veterinary education [17,18,19,20]. Thus, it is possible that this shift in focus to communication and collaboration, particularly with a focus on empathy, impacted perceptions of veterinarians as approachable, sensitive, sympathetic and understanding. Additionally, veterinarians help clients make decisions on the behalf of their animal that often necessitate a challenging deliberation regarding treatment options, quality of life, prognosis and end-of-life decision making in ways unparalleled in human medicine. Further complicating this decision making is the cost of medicine. A majority of people have some form of health insurance for themselves and their family members, but only about 1% of cat and dog owners in the US have pet insurance [21], making cost a very real constraint in the decision-making process. As a result, humane euthanasia is sometimes an option when recommended treatments are unaffordable. Beyond affordability, veterinarians also have euthanasia as an option to provide relief from pain and suffering in cases of terminal illness. These unique aspects of veterinary medicine necessitate a patient and compassionate approach to care when clients are most vulnerable. 

A second reason for the difference in perceptions might relate to television and media influences. In addition to the aforementioned qualities, veterinarians were perceived as slightly more helpful, likable, respectful and attentive than physicians. Previous research suggested that fiction doctors during prime-time shows were depicted as mean and unethical (e.g., House) [22], though more recent television shows have broadened the ways doctors are depicted. While fiction portrayals of veterinarians are limited, nonfiction such as reality television shows about veterinarians have been on the rise [23]. While we have no specific data to suggest that our sample viewed television programs about doctors or veterinarians, we speculate that perhaps media influences may be at play. A recent article published by the American Veterinary Medical Association asserted that veterinarians interviewed about reality programming “see the shows as a powerful medium to teach audiences about the profession they love while showcasing their medical skills and the importance of caring for animals” [23]. It is therefore possible that regular viewers of these reality shows would develop perceptions of veterinarians that are consistent with the image portrayed [24,25]. 

Third, at a basic level, we surmise that veterinarians’ very profession—that of healing and helping animals—impacts perceptions of veterinarians as caring professionals, likely reinforced through the aforementioned media representations and personal experience, which could translate to slightly more favorable ratings overall. Conversely, if we look closely at the concepts where physicians were ranked higher (confident, proud, inefficient, overconfident, arrogant, unpleasant and greedy) a pattern emerges related to public perceptions about physicians’ sense of self-importance. Again, possible explanations include personal experience with both groups of professionals and perceptions of roles based on television and other media. It is also possible that the bureaucracies associated with the health insurance industry and the commodification of human medicine in the US influenced these perceptions, particularly that physicians are inefficient and greedy. 

Finally, and perhaps most importantly, it appeared that public perceptions of veterinarians were in fact favorable, despite veterinary clients’ decreasing positive perceptions of veterinary hospitals and previous trends that have suggested a loss of confidence in the veterinary profession as a whole [4]. These results suggest that both the pet owning public and the non-pet-owning public are able to decouple the perceptions of veterinarians from the hospitals that employ them, perhaps due to a general perception of hospitals as businesses with financial motives.

While these results point to positive public perceptions of veterinarians, we acknowledge that this study was not without limitations. Although this study involved a national sample of survey participants and post-stratification weights were used to make the data mostly representative, the study did possess some limitations with respect to some racial/ethnic and age variables. More specifically, American Indian and Alaskan Natives (n = seven) and persons older than 65 (n = 26) were under-represented. We believe this sampling issue likely was an artifact of using Amazon mTurk for acquiring survey participants. Second, we did not collect information on whether our respondents owned animals or if they visited a veterinarian or physician in the past, nor did we ask about the quality of their prior experiences. Respondents’ perceptions were likely influenced by prior experiences with animals, veterinarians and physicians. Finally, while differences did exist, we used a nine-point rating scale to better discriminate responses. So, while veterinarians were rated more favorably, we acknowledge that differences were slightly given where the actual mean values fell along that nine-point continuum.

## 5. Conclusions

In closing, we see the key educational implications of this research. These results point to the favorable public perceptions of veterinarians in the US—a point that should be emphasized to current and prospective students. This is especially noteworthy given the veterinary students’ strong desire for respect as a core value [5]. It would appear that among the general public, they are, in fact, respected. Reinforcing the public’s strong trust in the veterinary profession throughout students’ education could enhance their own self-concept, self-esteem and overall mental health and well-being. Moreover, with this trust comes responsibilities. A second implication is the importance of instilling in students a belief in the power of their leadership. Specifically, veterinarians were viewed as caring, ethical and honest, all attributes associated with effective leadership [26]. To be a member of a profession that is bestowed with trust and credibility also requires a level of social responsibility to the public. Part of the responsibility includes increasing public awareness of the important contributions of veterinary medicine to animal, human, and environmental health. As a result, we recommend instilling in students a sense of the importance of public contact as a way to further enhance public perceptions and veterinary medicine’s professional image and expand the public understanding of the role of veterinarians in One Health. 

## Figures and Tables

**Table 1 vetsci-07-00050-t001:** Demographic characteristics of the sample (unweighted).

	N	%
Gender		
Male	302	49.8
Female	301	49.7
Other	3	0.5
Race		
White/Caucasian	467	77.1
Black or African American	55	9.1
American Indian and Alaskan Native	7	1.2
Asian	54	8.9
Other	3	0.5
Missing	20	3.3
Age ranges		
18–34	323	53.3
35–49	167	27.6
50–64	90	14.9
>64	26	4.3
Geographic region		
Northeast	138	22.8
Midwest	122	20.1
South	209	34.5
West	137	22.6
Highest education obtained		
Less than high school diploma	5	0.8
High school graduate	63	10.4
Some college, no degree	113	18.6
Associate’s degree (AS, AA, AA&S, etc.)	52	8.6
Bachelor’s degree (BS, BA, etc.)	272	44.9
Master’s degree (MS, MA, MFA, MBA, etc.)	79	13.0
Doctoral or Professional degree (PhD, MD, PharmD, JD, etc.)	22	3.6

**Table 2 vetsci-07-00050-t002:** Comparison of the ratings for physicians and veterinarians.

	Physicians		Veterinarians					
	Unweighted (n = 606)	Weighted (n = 758)	Unweighted (n = 606)	Weighted (n = 758)				
	Mean (SD)	Mean (SD)	Mean (SD)	Mean (SD)	*t*	df	*p*	*d*
Proud	7.30 (1.53)	7.32 (1.57)	6.93 (1.70)	6.92 (1.71)	7.107	757	<0.001	1.141
Arrogant	4.89 (2.35)	4.88 (2.35)	3.35 (2.25)	3.01 (2.11)	22.357	757	<0.001	0.838
Sensitive	5.59 (2.01)	5.37 (2.04)	6.89 (1.92)	6.98 (1.83)	−20.62	757	<0.001	0.831
Sympathetic	6.36 (1.82)	6.12 (1.80)	7.39 (1.65)	7.50 (1.54)	−19.659	757	<0.001	0.825
Approachable	6.33 (1.71)	6.14 (1.63)	7.28 (1.55)	7.40 (1.44)	−20.536	757	<0.001	0.819
Patient	5.68 (2.04)	5.47 (2.02)	6.84 (1.83)	6.94 (1.85)	−17.398	757	<0.001	0.760
Overconfident	5.15 (2.36)	5.06 (2.34)	3.77 (2.35)	3.38 (2.20)	20.986	757	<0.001	0.740
Understanding	6.40 (1.84)	6.29 (1.74)	7.33 (1.66)	7.39 (1.54)	−16.868	757	<0.001	0.670
Punctual	5.54 (2.17)	5.43 (2.15)	6.63 (1.68)	6.70 (1.62)	−16.820	757	<0.001	0.667
Likable	6.51 (1.64)	6.39 (1.62)	7.28 (1.57)	7.39 (1.46)	−16.714	757	<0.001	0.649
Respectful	6.68 (1.70)	6.52 (1.71)	7.42 (1.55)	7.50 (1.44)	−15.657	757	<0.001	0.620
Caring	6.72 (1.66)	6.59 (1.70)	7.42 (1.62)	7.55 (1.49)	−15.908	757	<0.001	0.599
Attentive	6.69 (1.73)	6.52 (1.73)	7.38 (1.62)	7.42 (1.58)	−14.410	757	<0.001	0.543
Helpful	6.72 (1.74)	6.63 (1.68)	7.41 (1.57)	7.45 (1.56)	−13.045	757	<0.001	0.505
Unpleasant	3.83 (2.23)	3.70 (2.13)	2.99 (2.23)	2.69 (2.07)	14.933	757	<0.001	0.480
Thorough	6.67 (1.75)	6.69 (1.74)	7.25 (1.63)	7.39 (1.54)	−12.344	757	<0.001	0.427
Greedy	4.33 (2.32)	4.15 (2.28)	3.65 (2.41)	3.27 (2.33)	11.718	757	<0.001	0.382
Ethical	6.83 (1.71)	6.77 (1.70)	7.29 (1.62)	7.39 (1.54)	−9.975	757	<0.001	0.382
Honest	6.92 (1.51)	6.82 (1.53)	7.13 (1.57)	7.21 (1.50)	−7.397	757	<0.001	0.257
Competent	7.06 (1.60)	7.02 (1.58)	7.34 (1.54)	7.41 (1.49)	−7.500	757	<0.001	0.253
Inefficient	3.71 (2.15)	3.45 (2.05)	3.30 (2.21)	3.01 (2.05)	7.160	757	<0.001	0.215
Confident	7.45 (1.51)	7.48 (1.47)	7.19 (1.56)	7.22 (1.50)	5.333	757	<0.001	0.175
Rational	7.00 (1.52)	7.00 (1.52)	7.17 (1.62)	7.26 (1.55)	−5.157	757	<0.001	0.170
Skilled	7.29 (1.56)	7.32 (1.52)	7.43 (1.52)	7.54 (1.48)	−4.235	757	<0.001	0.146
Scientific	7.05 (1.63)	7.07 (1.61)	7.00 (1.71)	7.10 (1.65)	−0.550	757	0.583	0.018

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
