# Peer review of "A Comparison of Public Perceptions of Physicians and Veterinarians in the United States"

_vetsci, 2020, doi:10.3390/vetsci7020050_

Round 1
Reviewer 1 Report
A timely piece that is certain to be of interest to the readership of Veterinary Sciences and to the public health community as a whole. This paper sets the stage for future research related to the impacts of the findings in the current study. Well done. Recommend publishing in its current form.
Author Response
Thank you for your supportive comments about our manuscript.
Reviewer 2 Report
I enjoyed reading 'A comparison of public perceptions of physicians and veterinarians'. My recommendation is that the manuscript is acceptable for publication following minor amendments based on the following comments:
Line 2-3 The title should reference the US (see comments below)
Line 4. The manuscript only lists a single author, whereas the journal information lists two authors.
Line 13. Can the sample be referred to as nationally representative when there is an issue with diversity? 80% of respondents were < 49 years old of which 53% were between 18 and 34.
Lines 24-62 Whilst the introduction is well-written, there is a greater body of literature that could be cited e.g. papers on professionalism in veterinary education etc. The Vet Futures project in the UK reported in 2015 on public trust in healthcare professionals including vets. https://www.vetfutures.org.uk/resource/public-trust-in-the-professions-may-2015/ In addition, there are public polls from the UK regarding trust in physicians https://www.ipsos.com/ipsos-mori/en-uk/public-trust-doctors-remains-high
Both of the above links are UK-based and it is likely that there is a cultural component. There are papers to demonstrate that public perceptions towards health care depends on whether it is commodified. It may be appropriate for the manuscript to provide a clear context - that it is based on the US. This should be reflected in the title, discussion and the conclusions.
Regarding the discussion - the use of mTurk and understanding the demographic of the participants requires further discussion/analysis. For example, perceptions will be influenced by experience (Line 148), which was not explored/presented. It would be beneficial to examine the survey responses according to participants with a recent experience of either a vet or a physician and the quality of their experience. Did you record additional details about the participants e.g. animal ownership? If not this should be noted as a limitation to the scope of the research, to ensure that the results are not over-interpreted.
Line 159-160 Physicians also provide palliative care
Line 175 To avoid speculating, it would be better to provide arguments supported by the literature.
Line 182-186 Is it appropriate to make this comparison? The reference cited [4] refers to veterinary clients. Whereas in the current study we cannot differentiate between animal owners and non-owners.
One of the conclusions stated in the abstract (lines 18-19) is that "These findings are particularly relevant for veterinary educators who train the future workforce and have a significant role in how the profession is portrayed." This differs slightly to the conclusions (lines 197-207), which provides an important message about public trust in the veterinary profession and social responsibility. Veterinary educators support the development of competency, one of which is professional growth - ethics, communication etc. The reputation and trust of the veterinary profession is underpinned by professionalism such as ethical values and behaviour.
Line 185 spelling error
Author Response
Line 2-3 The title should reference the US (see comments below)
Response: We have added US to the title to reflect that this research was conducted in the US.
Line 4. The manuscript only lists a single author, whereas the journal information lists two authors.
Response: There are two authors and that has been corrected.
Line 13. Can the sample be referred to as nationally representative when there is an issue with diversity? 80% of respondents were < 49 years old of which 53% were between 18 and 34.
Response: The sample was mostly representative. We made this change within the manuscript.
Lines 24-62 Whilst the introduction is well-written, there is a greater body of literature that could be cited e.g. papers on professionalism in veterinary education etc. The Vet Futures project in the UK reported in 2015 on public trust in healthcare professionals including vets. https://www.vetfutures.org.uk/resource/public-trust-in-the-professions-may-2015/ In addition, there are public polls from the UK regarding trust in physicians https://www.ipsos.com/ipsos-mori/en-uk/public-trust-doctors-remains-high
Both of the above links are UK-based and it is likely that there is a cultural component. There are papers to demonstrate that public perceptions towards health care depends on whether it is commodified. It may be appropriate for the manuscript to provide a clear context - that it is based on the US. This should be reflected in the title, discussion and the conclusions.
Response: We added clarification that this study relates to the United States in the title, discussion, and conclusion and also addressed the commodification of human medicine in the discussion.
Regarding the discussion - the use of mTurk and understanding the demographic of the participants requires further discussion/analysis. For example, perceptions will be influenced by experience (Line 148), which was not explored/presented. It would be beneficial to examine the survey responses according to participants with a recent experience of either a vet or a physician and the quality of their experience. Did you record additional details about the participants e.g. animal ownership? If not this should be noted as a limitation to the scope of the research, to ensure that the results are not over-interpreted.
Response: We did not collect information about animal ownership and have acknowledged this as a limitation in the manuscript.
Line 159-160 Physicians also provide palliative care
Response: We agree that doctors also provide palliative care, but the ability to end suffering via euthanasia is unique to the veterinary context. We have clarified this.
Line 175 To avoid speculating, it would be better to provide arguments supported by the literature.
Response: While we could find no specific research linking healing animals to perceptions of caring independent of client’s expectations and/or burnout and compassion fatigue, we did link back to media representations and personal experience.
Line 182-186 Is it appropriate to make this comparison? The reference cited [4] refers to veterinary clients. Whereas in the current study we cannot differentiate between animal owners and non-owners.
Response: We agree that we do not know the percentage of our sample who have/had pets and this have experience with veterinarians. We’ve clarified this in the discussion to make this transparent.
One of the conclusions stated in the abstract (lines 18-19) is that "These findings are particularly relevant for veterinary educators who train the future workforce and have a significant role in how the profession is portrayed." This differs slightly to the conclusions (lines 197-207), which provides an important message about public trust in the veterinary profession and social responsibility. Veterinary educators support the development of competency, one of which is professional growth - ethics, communication etc. The reputation and trust of the veterinary profession is underpinned by professionalism such as ethical values and behaviour.
Response: We added information to the abstract to be consistent with the conclusions about public trust and social responsibility.
Line 185 spelling error
Response: Thank you! We fixed it.

Reviewer 3 Report
Brief summary – The submitted paper describes results from a 9 point Likert-scale survey on the public perception of a representative sample of the US population regarding physicians and vets. The paper is well written and is of interest to the readers of Veterinary Sciences. Yet, some issues require clarification and further reflection.
Broad Comments – Thank you for submitting such an interesting paper. The introduction gives a clear idea of the remit of your research but I think that the authors can strengthen the relation between physicians and veterinarians by alluding the concept of One Health. In fact, results seem to indicate that physicians would benefit from being compared to vets. I think that the discussion needs to be improved. In the introduction and conclusion, you mention the power of leadership but you don’t discuss how the concepts that you have explored relate to being a good leader (e.g. https://veterinaryrecord.bmj.com/content/182/16/460) or a successful veterinarian (or indeed an ethical veterinarian). In the discussion, please describe how your results can be framed within the One Health concept.
I also have some specific comments that will hopefully improve the paper.
Specific Comments
Title: Where was the survey conducted?
Abstract:
Line 13 – Nationally? You mean US
Line 13 – 606 or 758? Shouldn’t you refer to the sample after post-stratification weights?
Introduction:
Please introduce the concept of One Health.
Line 25 – Where was the 1999 study conducted?
Line 27 – What about physicians? How were they ranked? Did they rank first in any category?
Line 30 – Please explain in further detail how pet owners’ perceptions about veterinary hospitals have decreased.
Line 37 – DMV. In full or use alternative.
Line 46 – “Further evidence suggests that the public may be losing confidence in veterinarians, despite the 2006 Gallup study results.” This sentence needs to be illustrated by examples. I suggest:
It shouldn’t Happen at a Vets’, Panorama, BBC1, 22-07-2010;
Pedigree Dogs Exposed, BBC1, 19-08-2008;
Pedigree Dogs Exposed – Three Years On, BBC4, 27-02-2012
Methods:
Line 67 – 3.18% - How was this figure reached?
Table 1 – Shouldn’t you compare your sample with the demographic characteristic of the US population?
Line 78 – You say to have chosen the words arbitrarily but also say that they were chosen for being particularly salient. Doesn’t this mean that the words were chosen purposefully (rather than arbitrarily)?
Line 79 – You have used favorable, unfavorable and neutral words. Please provide a table with the words organised in such categories.
Line 96 – I take from your words that physicians’ questions always came first. Why haven’t you randomised the ordering? Please explain in the discussion.
Line 115- Which University? What is the reference number for the exemption?
Results:
Table 2 – How were the terms organised? Is it by order of appearance in the survey? I suggest a more intuitive ordering by d values. In the description you should also clarify that results with negative t test indicate a more favourable perception towards vets and that results with positive t test indicate a more favourable perception towards physicians.
Discussion:
Line 144 - I think that the public tends to perceive veterinarians more than just “slightly more favorably than physicians”. Vets score better at any positive/neutral concept, except for Confident and Proud, and score worse for the negative concepts. Although differences can be small they are consistent and statistically ‘significant’ (although it is known that significance does not mean that a given hypothesis is correct). Taken together, results seem to indicate that vets are indeed perceived more favourably by the public.
Line 156-157 – “A majority of people have some form of health insurance for themselves and their family 156 members, but only about 1% of cat and dog owners have pet insurance” Do you mean in the US? Where does the 1% come from?
Line 177 - Only a few lines were used to explain the physicians’ concepts. More needs to be said about the roles and qualities of a good physician.
Line 185 – separate (typo)
Line 190 – You say that “The study does possess some limitations with respect to some racial/ethnic and age variables” but we don’t have access to the demographics of the US population.
Author Response
Brief summary – The submitted paper describes results from a 9 point Likert-scale survey on the public perception of a representative sample of the US population regarding physicians and vets. The paper is well written and is of interest to the readers of Veterinary Sciences. Yet, some issues require clarification and further reflection.
Broad Comments – Thank you for submitting such an interesting paper. The introduction gives a clear idea of the remit of your research but I think that the authors can strengthen the relation between physicians and veterinarians by alluding the concept of One Health. In fact, results seem to indicate that physicians would benefit from being compared to vets. I think that the discussion needs to be improved. In the introduction and conclusion, you mention the power of leadership but you don’t discuss how the concepts that you have explored relate to being a good leader (e.g. https://veterinaryrecord.bmj.com/content/182/16/460) or a successful veterinarian (or indeed an ethical veterinarian). In the discussion, please describe how your results can be framed within the One Health concept.
Response: We added One Health to the Introduction and Conclusion.
I also have some specific comments that will hopefully improve the paper.
Specific Comments
Title: Where was the survey conducted?
Response: We conducted the survey in the US and have added this to the title.
Abstract:
Line 13 – Nationally? You mean US
Response: Yes, we have clarified this in the abstract.
Line 13 – 606 or 758? Shouldn’t you refer to the sample after post-stratification weights?
Response: We explained 606 participants (unweighted) completed the survey.
Introduction:
Please introduce the concept of One Health.
Response: We have added One Health to the introduction.
Line 25 – Where was the 1999 study conducted?
Response: This study was conducted in the United States. We added this information to the introduction.
Line 27 – What about physicians? How were they ranked? Did they rank first in any category?
Response: Physicians also ranked favorably, as explained in the article, occupying the top 3 spots for most categories. This information came from a bound collection of JAVMA journals that was accessed via the NC State library. Due to COVID-19 the library is closed for the foreseeable future, so I am unable to retrieve the journal to further reference the article to determine the exact categories in which physicians ranked first. However, we do report rankings of physicians in the later Gallup polls.
Line 30 – Please explain in further detail how pet owners’ perceptions about veterinary hospitals have decreased.
Response: We added additional detail to explain the decrease in perceptions.
Line 37 – DMV. In full or use alternative.
Response: We specified veterinary medicine and removed DVM.
Line 46 – “Further evidence suggests that the public may be losing confidence in veterinarians, despite the 2006 Gallup study results.” This sentence needs to be illustrated by examples. I suggest:
Response: We included two specific examples from US media as evidence.
Methods:
Line 67 – 3.18% - How was this figure reached?
Response: We calculated the margin of error using an online margin of error calculator.
Table 1 – Shouldn’t you compare your sample with the demographic characteristic of the US population?
Response: We did not find it necessary to do so.
Line 78 – You say to have chosen the words arbitrarily but also say that they were chosen for being particularly salient. Doesn’t this mean that the words were chosen purposefully (rather than arbitrarily)?
Response: We have removed the word “arbitrarily” as our selections were not completely random.
Line 79 – You have used favorable, unfavorable and neutral words. Please provide a table with the words organised in such categories.
Response: We have opted not to include another table listing the words from favorable to unfavorable, as this is intuitive for most words, and for others, this could be subjective. We attempted to include a diverse mix of words and simply wanted to explore what, if any, differences in perception exist.
Line 96 – I take from your words that physicians’ questions always came first. Why haven’t you randomised the ordering? Please explain in the discussion.
Line 115- Which University? What is the reference number for the exemption?
Response: NC State University IRB approved this research (protocol # 15528), which we added to the manuscript.
Results:
Table 2 – How were the terms organised? Is it by order of appearance in the survey? I suggest a more intuitive ordering by d values. In the description you should also clarify that results with negative t test indicate a more favourable perception towards vets and that results with positive t test indicate a more favourable perception towards physicians.
Discussion:
Line 144 - I think that the public tends to perceive veterinarians more than just “slightly more favorably than physicians”. Vets score better at any positive/neutral concept, except for Confident and Proud, and score worse for the negative concepts. Although differences can be small they are consistent and statistically ‘significant’ (although it is known that significance does not mean that a given hypothesis is correct). Taken together, results seem to indicate that vets are indeed perceived more favourably by the public.
Response: We did not want to overstate our findings so included the word “slightly.” We have since removed it.
Line 156-157 – “A majority of people have some form of health insurance for themselves and their family 156 members, but only about 1% of cat and dog owners have pet insurance” Do you mean in the US? Where does the 1% come from?
Response: Yes, in the US. We added this in for clarification and included a reference.
Line 177 - Only a few lines were used to explain the physicians’ concepts. More needs to be said about the roles and qualities of a good physician.
Response: We added additional information in the discussion to unpack the characteristics were physicians were rated higher.
Line 185 – separate (typo)
Response: We fixed it. Thank you!
Line 190 – You say that “The study does possess some limitations with respect to some racial/ethnic and age variables” but we don’t have access to the demographics of the US population.
Response: We state which demographic groups were underrepresented in our sample in the limitations.

Reviewer 4 Report
This study is interesting, written clearly, and the methodology is sound as as well as the discussion.
Author Response
Thank you for the supportive feedback on our manuscript.